

# Characterization of reactive nitrogen in the global upper troposphere using recent and historic commercial and research aircraft campaigns and GEOS-Chem

Nana Wei[1], Eloise A. Marais[1], Gongda Lu[1*], Robert G. Ryan[1**], Bastien Sauvage[2]

[1] Department of Geography, University College London, London, UK.
[2] Laboratoire d'Aérologie, Université de Toulouse, CNRS, Université Toulouse III Paul Sabatier, France.
[*] Now at: the Satellite Application Centre for Ecology and Environment (SACEE), Beijing, China.
[**] Now at: School of Geography, Earth and Atmospheric Science, University of Melbourne, Melbourne,
Australia.

*Correspondence to*: Nana Wei (nana.wei.21@ucl.ac.uk); Eloise A. Marais (e.marais@ucl.ac.uk)

**Abstract.** Reactive nitrogen ($NO_y$) in the upper troposphere (UT; ~8-12km) influences global climate, air quality, and tropospheric oxidants, but this is informed by limited knowledge of the relative contribution of individual $NO_y$ components in this undersampled layer. Here we use sporadic NASA DC-8 aircraft campaign observations,

after screening for plumes and stratospheric influence, to characterise UT $NO_y$ composition and evaluate current knowledge of UT $NO_y$ as simulated with the GEOS-Chem model. Use of DC-8 data follows confirmation that these sporadic data reproduce $NO_y$ seasonality from routine commercial aircraft observations (2003-2019), supporting use of DC-8 data to characterize UT $NO_y$. We find that peroxyacetyl nitrate (PAN) dominates UT $NO_y$ (30-64% of $NO_y$), followed by nitrogen oxides ($NO_x \equiv NO + NO_2$) (6-18%), peroxynitric acid ($HNO_4$) (6-13%),

and nitric acid ($HNO_3$) (7-11%). Methyl peroxy nitrate (MPN) makes an outsized contribution to $NO_y$ (24%) over the Southeast US relative to the other regions sampled (2-7%). GEOS-Chem, sampled along DC-8 flights, exhibits much weaker seasonality than DC-8, underestimating summer and spring $NO_y$ and overestimating winter and autumn $NO_y$. The model consistently overestimates peroxypropionyl nitrate (PPN) by up to 16 pptv and underestimates $NO_2$ by 6-36 pptv, as the model is missing PPN photolysis. An ~80 pptv (20-fold) underestimate

in modelled MPN over the Southeast US results from uncertainties in processes that sustain MPN production as air ages. Our findings highlight that greater understanding of UT $NO_y$ is critically needed to determine its role in the nitrogen cycle, air pollution, climate, and abundance of oxidants.

## 1 Introduction

Reactive nitrogen ($NO_y$) in the upper troposphere (~8-12 km) impacts global climate, surface air quality and the

oxidizing capacity of the whole troposphere (Bradshaw et al., 2000; Dahlmann et al., 2011; Mickley et al., 1999; Worden et al., 2011). $NO_y$ is an important climate driver because tropospheric ozone ($O_3$) production is limited by the availability of $NO_y$, particularly in the upper troposphere where the radiative forcing efficiency of $O_3$ peaks (Dahlmann et al., 2011; Rap et al., 2015; Worden et al., 2011). Influence on tropospheric $O_3$ production also affects abundance of the main atmospheric oxidant, the hydroxyl radical (OH), thus altering the lifetimes of the

longer-lived greenhouse gas methane and the air pollutants carbon monoxide (CO) and volatile organic compounds (VOCs) (Murray et al., 2013; Seltzer et al., 2015).



Knowledge of dominant daytime $NO_y$ compounds, sources, chemistry, fate, and persistence in the upper troposphere has been largely informed by observations and models used as part of research and commercial aircraft campaigns (Boersma et al., 2011; Marais et al., 2018; Silvern et al., 2018; Travis et al., 2020; Travis et al.,

2016). Instruments onboard research aircraft that sample the upper troposphere, in particular the recently retired NASA DC-8 platform, have undergone substantial development to directly measure and derive estimates of a large suite of upper tropospheric $NO_y$ compounds. These include nitrogen oxides ($NO_x \equiv NO + NO_2$), peroxyacetyl nitrate (PAN) and other prominent PAN-type compounds, nitric acid ($HNO_3$), peroxynitric acid ($HNO_4$), alkylnitrates (ALKNs) and, more recently, methyl peroxy nitrate (MPN).

These aircraft campaigns have confirmed that sources of $NO_y$ to the upper troposphere are dominated by lightning $NO_x$ emissions (Marais et al., 2018; Levy Ii et al., 1999; Gressent et al., 2016; Gressent et al., 2014), causing a seasonal maximum in $NO_y$ in summer months and a minimum in winter in parts of the world such as the northern midlatitudes where there is large seasonal variability in lightning activity (Stratmann et al., 2016; Blakeslee et al., 2014). Other $NO_y$ source contributors include $NO_x$ emissions from cruising altitude aircraft (Brasseur et al., 1996),

stratospheric downwelling of air masses laden with $HNO_3$ and $NO_2$ that also promote prompt formation of PANs on mixing with cold upper tropospheric air (Liang et al., 2011; Jacob et al., 2010; Levy Ii et al., 1980), deep convective uplift of surface pollution (Ehhalt et al., 1992; Jaeglé et al., 1998; Bertram et al., 2007), and aged air masses initially very photochemically active that accumulate MPN (Nault et al., 2015).

Chemical cycling of dominant daytime $NO_y$ components in the upper troposphere is illustrated in Figure 1. During

the day, NO and $NO_2$ are in photostationary steady state, as NO oxidation, mostly by $O_3$, is balanced by $NO_2$ photolysis. $NO_x$ also reacts to form reservoir compounds. For $NO_2$, these include $HNO_3$ from reaction with OH, PANs from reaction with peroxy acyl radicals (RC(O)OO), $HNO_4$ from reaction with the hydroperoxyl radical ($HO_2$), and MPN from reaction with the methyl peroxy radical ($CH_3O_2$). PANs in the upper troposphere are typically dominated by PAN followed by peroxypropionyl nitrate (PPN) (Roberts, 1990; Roberts et al., 1998;

Roberts et al., 2002; Singh, 1987). For NO, reservoir compounds include ALKNs from reaction with non-acyl peroxy radicals ($RO_2$). Recycling of reservoir compounds back to $NO_x$ is dominated by photolysis, as thermally labile peroxy nitrates (PNs) including PANs, $HNO_4$ and MPN are stable against decomposition in the cold upper troposphere. This recycling along with $NO_y$ sources to the upper troposphere sustains upper tropospheric $NO_x$ concentrations at ~30 pptv over the remote ocean and ~100 pptv over polluted landmasses (Shah et al., 2023;

Marais et al., 2018; Marais et al., 2021). Stable $NO_x$ reservoir compounds are transported long distances before subsiding and decomposing on warming, thus supplying other parts of the world with oxidants ($HO_x$) and $O_3$ precursors ($NO_x$ and peroxy radicals). Loss processes in the dry upper troposphere are slow and dominated by subsidence. In the upper troposphere, $NO_y$ has a lifetime of 10-20 days (Logan, 1983; Prather and Jacob, 1997) and $NO_x$ has a lifetime of about a week compared to less than a day in the boundary layer (<2 km) (Jaeglé et al.,

70    1998).



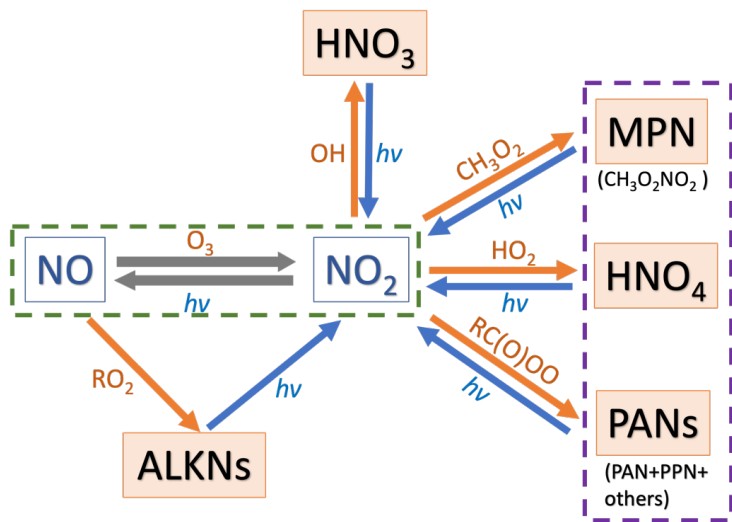

**Figure 1: Dominant daytime reactive nitrogen components and reaction pathways in the upper troposphere. Arrow colours distinguish formation (orange) and photolytic (hv) decomposition (blue) of reservoir compounds. Dashed boxes indicate compounds of the $NO_x$ family (green) and classed as peroxy nitrates (purple). "R" in RC(O)OO and $RO_2$ represents an alkyl group.**

Modelling studies evaluating best understanding of $NO_y$ in the upper troposphere routinely identify stark discrepancies between observed and modelled total $NO_y$, $NO_x$, and the ratio of NO-to-$NO_2$ in the upper layers of the troposphere. These studies have either focused on a few $NO_y$ components, or a single aircraft campaign (Talbot et al., 1999; Lee et al., 2022; Bertram et al., 2007; Huntrieser et al., 2016; Liang et al., 2011; Nault et al., 2015; Fisher et al., 2018; Cohen et al., 2023). A more holistic investigation of all $NO_y$ components is needed, as is advocated by Murray et al. (2021), to reduce uncertainties in knowledge of the current, past, and potential future abundances of tropospheric oxidants. Past studies have also documented the challenges examining measurements made in the upper troposphere. These include screening for stratospheric influence, determining the height of the chemical tropopause, and selecting observations and campaigns that are climatologically representative of a standard atmosphere (Barth et al., 2015; Fuelberg et al., 2000; Bertram et al., 2007; Huntrieser et al., 2016; Weinheimer et al., 1994). Instruments measuring $NO_2$ are also susceptible to interference from decomposition of the least thermally stable $NO_x$ reservoir compounds, $HNO_4$ and MPN, that are abundant in the cold upper troposphere (Shah et al., 2023; Ryerson et al., 2000). $NO_y$ from these same instruments can also be biased by decomposition of non-$NO_y$ fixed nitrogen compounds prevalent in the upper troposphere, such as hydrogen cyanide (HCN) (Bradshaw et al., 1998).

Here we use NASA DC-8 research and IAGOS commercial aircraft campaign measurements, each spanning more than a decade, to characterize global $NO_y$ seasonality and composition in the upper troposphere. This follows careful campaign and data selection to isolate observations sampling the upper troposphere under standard conditions and broad assessment of consistent $NO_y$ seasonality between DC-8 and routine commercial aircraft campaign observations. We go on to use the DC-8 data to critique contemporary understanding of upper tropospheric $NO_y$ as simulated by the GEOS-Chem model.



## 2 Materials and methods

### 2.1 Research aircraft observations of total and components of $NO_y$

The DC-8 research aircraft has sampled ambient air covering the full extent of the troposphere since its maiden campaign in 1985 (Culter, 2009). Many of the initial campaigns included instruments that measured a subset of the $NO_y$ components shown in Figure 1, typically continuous measurements of total $NO_y$, NO, $HNO_3$, PAN and PPN, and whole air sampler (WAS) collection and laboratory detection of C1-C5 ALKNs (Singh et al., 1999). Since 2004, DC-8 campaigns have included continuous measurements of $HNO_4$, other PAN-type species and total

PNs. Given this, we only consider DC-8 campaigns with a relatively consistent suite of instruments that mostly sampled well-mixed air representative of a climatologically standard atmosphere and that have limited influence from stratospheric air. These criteria eliminate the summer 2004 Intercontinental Chemical Transport Experiment-North America (INTEX-NA) campaign (Singh et al., 2006; Singh et al., 2009) that is the only DC-8 campaign since 2004 to not include a $NO_x$ and $NO_y$ chemiluminescence analyzer, and the summer 2012 Deep Convective

Clouds and Chemistry (DC3) campaign that targeted convective thunderstorms influenced by fresh surface pollution and lightning $NO_x$ emissions (Barth et al., 2015).

The DC-8 campaigns we use are the Arctic Research of the Composition of the Troposphere from Aircraft and Satellites (ARCTAS) over the Arctic and sub-Arctic in spring and summer 2008 (Jacob et al., 2010), the Studies of Emissions and Atmospheric Composition, Clouds and Climate Coupling by Regional Surveys (SEAC[4]RS) over

the Southeast US in late summer and early autumn 2013 (Toon et al., 2016), the Korea-United States Air Quality (KORUS-AQ) over South Korea in late spring and early summer 2016 (Crawford et al., 2021), and the Atmospheric Tomography Mission (ATom) that included 4 sub-campaigns along the same flight path from pole to pole over the Atlantic and Pacific Oceans in all 4 seasons from 2016 to 2018 (Thompson et al., 2021). ATom sub-campaigns are ATom-1 in July-August, ATom-2 in January-February, ATom-3 in September-October and

ATom-4 in April-May. The data for these campaigns are from NASA data portals for each campaign downloaded as merged 1-minute files for ARCTAS (NASA, 2009), SEAC[4]RS (NASA, 2015) and KORUS-AQ (NASA, 2017) and as two separate merged files for ATom with the WAS C1-C5 ALKNs data at variable time intervals of 40 s, 1 min and 2 min and without the WAS C1-C5 ALKNs data at 1-minute resolution (NASA, 2021).

Figure 2 shows the global sampling extent of the upper troposphere by NASA DC-8 after applying filtering criteria

to the data to isolate observations representative of photochemical steady-state conditions. For this, we select daytime (08h30-15h30 local solar time or LST) observations within a wide pressure range of 180 to 450 hPa (~8-12 km) to cover the full vertical extent of the upper troposphere that varies with season and latitude. We separate the stratosphere from the troposphere with a tropopause definition that can be applied to all datasets. We remove data with observed $O_3$ concentrations above thresholds that represent the location of the chemical tropopause

(Zahn et al., 2002). The thresholds we use are a single year-round value for the tropics (20°N to 20°S) of 100 ppbv (Dameris, 2015) and seasonally varying values everywhere else calculated using the day-of-year dependent $O_3$ tropopause equation derived by Zahn et al. (2002) from the inverse relationship between $O_3$ and CO observations from commercial aircraft campaigns. These are 120 ppbv in spring, 103 ppbv in summer, 74 ppbv in autumn, and 91 ppbv in winter. We also screen for stratospheric intrusions (identified as observations with $O_3$/CO > 1.25 mol



mol$^{-1}$) (Hudman et al., 2007), fresh NO$_x$ emissions (NO$_y$/NO < 3 mol mol$^{-1}$), fresh convection (large (> 10 nm diameter) condensation nuclei > $10^4$ cm$^{-3}$), biomass burning plumes (CO > 200 ppbv and acetonitrile > 200 pptv) (Shah et al., 2023), as well as instances where NO$_2$ photolysis frequencies are approximately zero. The latter removes high latitude ATom measurements obtained at 08h30-15h30 LST under dark conditions during polar twilight or polar night. The data that are retained correspond to solar zenith angles ≤ $80^0$ in polar regions, and ≤

$60^0$ at other latitudes.

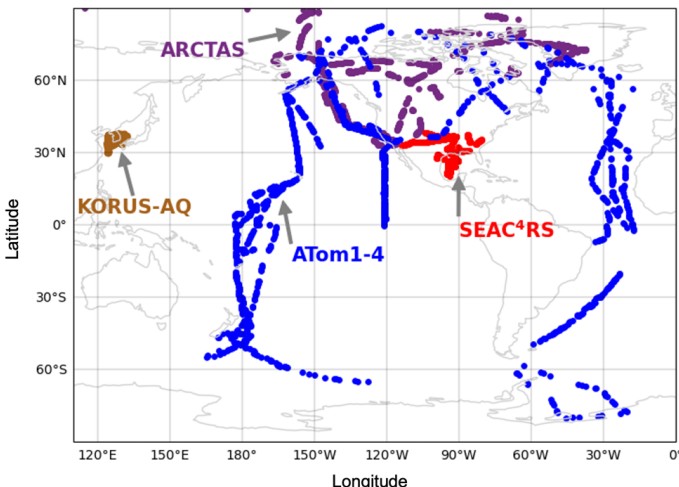

**Figure 2: Extent of NASA DC-8 sampling of the upper troposphere under standard, steady-state conditions. Colours distinguish ARCTAS (plum), SEAC$^4$RS (red), KORUS-AQ (brown), and ATom (blue). ATom**
**points are the 1-minute resolution data.**

The DC-8 instruments measuring NO$_y$ components (Figure 1) that are common to all campaigns include a chemiluminescence instrument measuring NO, NO$_2$, and total NO$_y$ (Ryerson et al., 2000; Pollack et al., 2010; Bourgeois et al., 2022), a chemical ionization mass spectrometer (CIMS) measuring HNO$_3$ (Crounse et al., 2006), a CIMS measuring HNO$_4$, PAN, PPN, and other PANs (Slusher et al., 2004), and a Whole Air Sampler (WAS)
collecting samples analysed in the laboratory using gas chromatography with flame ionization and atomic emission to detect C1-C5 ALKNs (Blake et al., 2003). The other PANs measured with the CIMS include peroxyacryloyl nitrate (APAN), peroxyisobutyryl nitrate (PiBN), peroxybutyryl nitrate (PBN), and peroxybenzoyl nitrate (PBZN). Other instruments deployed for select campaigns are Thermal-Dissociation Laser Induced Fluorescence (TD-LIF) measuring NO$_2$, total PNs and total ALKNs (ARCTAS, KORUS-AQ, SEAC$^4$RS) and the
PAN and Trace Hydrohalocarbon ExpeRiment (PANTHER) instrument measuring PAN (ATom). There are also TD-LIF methyl peroxy nitrate (MPN) measurements reported in the SEAC$^4$RS dataset and derived for ARCTAS by Browne et al. (2011).

Concentrations of NO$_2$ in the upper troposphere are close to chemiluminescence instrument uncertainty (Pollack
et al., 2010; Bourgeois et al., 2022) and the measurements include interference from decomposition of NO$_x$



reservoir compounds in the instrument inlet. The Reed et al. (2016) temperature-dependent inlet temperature decomposition profiles of individual $NO_x$ reservoir compounds for an instrument similar to that operated on the DC-8 suggests interference of 80-100% MPN and 15-45% $HNO_4$ for the typical inlet temperature range of the DC-8 chemiluminescence instrument of 20-30°C (Bourgeois et al., 2022). For the campaigns that measured $HNO_4$

and derived or measured MPN, this amounts to 13-27 pptv for ARCTAS and 71-92 pptv for SEAC⁴RS. Given this, we instead calculate $NO_2$ using the $NO$-$NO_2$ photochemical steady state (PSS) approximation, as is now standard (Horner et al., 2024; Shah et al., 2023; Travis et al., 2016). Conversion of NO to $NO_2$, mostly (75%) due to oxidation by $O_3$ in the upper troposphere (Silvern et al., 2018) , is balanced by $NO_2$ photolysis back to NO:

$$NO \xrightarrow{O_3/HO_2/BrO} NO_2 \qquad (R1)$$
$$NO_2 \xrightarrow{h\nu} NO \qquad (R2).$$

As $NO_x$ is in steady state for the daylight observations we isolate, $NO_2$ can be calculated as follows:

$$NO_2 = NO \times \left( \frac{k_1[O_3]+k_2[HO_2]+k_3[BrO]}{j_{NO_2}} \right) \qquad (1).$$

Compounds in square brackets are in molecules $cm^{-3}$. NO and $NO_2$ are in pptv. Terms not introduced yet include the $NO_2$ photolysis frequency, $j_{NO_2}$, in $s^{-1}$ and bromine monoxide (BrO), and rate constants of NO oxidation (R1) ($k_{1-3}$), are in $cm^3$ $molecule^{-1}$ $s^{-1}$. Temperature-dependent values of $k_{1-3}$ are those recommended by the Jet

Propulsion Laboratory (JPL) (Burkholder, 2020), calculated using DC-8 ambient temperature measurements. NO, $[O_3]$, and $j_{NO_2}$ are from the DC-8 measurements and $[HO_2]$ is from the DC-8 measurements for all campaigns, except SEAC⁴RS when it was not measured. We use GEOS-Chem (detailed in Sect. 2.3) simulated $[HO_2]$ to estimate SEAC⁴RS PSS $NO_2$. [BrO] is from GEOS-Chem for all campaigns. NO is also converted to $NO_2$ by organic peroxy radicals ($RO_2$), but we ignore this reaction as it is relatively insignificant in the upper troposphere

(Shah et al., 2023).

## 2.2 Commercial aircraft observations of total $NO_y$

We use routine observations of upper tropospheric total $NO_y$ from instruments on commercial long-haul passenger

aircraft to determine if the intermittency and brevity of DC-8 campaign observations are representative of climatological conditions. The In-service Aircraft for a Global Observing System (IAGOS) European research infrastructure (https://www.iagos.org, last accessed May 2024) provides routine in situ measurements of $NO_y$ (Petzold et al., 2015). These are available from two IAGOS programmes: the Measurement of Ozone and Water Vapor by Airbus In-Service Aircraft (MOZAIC) (Marenco et al., 1998) from 2001 to 2005 (Volz-Thomas et al.,

2005) and the Civil Aircraft for the Regular Investigation of the Atmosphere Based on an Instrument Container (CARIBIC) since December 2004 (Stratmann et al., 2016; Brenninkmeijer et al., 2007).

We consider the MOZAIC and CARIBIC observations together (collectively named IAGOS), as both programmes employed a chemiluminescence instrument with the same $NO_y$ detection technique (Brenninkmeijer et al., 2007;



Volz-Thomas et al., 2005). Direct intercomparison of $NO_y$ is not possible, as there is no overlap in MOZAIC and CARIBIC $NO_y$. Data from 2003 to 2019 are used; 2003-2005 for MOZAIC and 2005-2019 for CARIBIC. We isolate upper tropospheric observations by applying the same $O_3$ tropopause, stratospheric $O_3$ intrusion, and daytime filtering as is applied to DC-8 data (Sect. 2.1). We do not screen for observations impacted by fresh emissions, vertical convection or biomass burning plumes, due to unavailability of concurrent measurements of

suitable chemical tracers in the IAGOS data. As we consider 17 years of IAGOS data, we assume that the influence of these is dampened in the long-term median of $NO_y$. Both the IAGOS and DC-8 data are gridded to the same 2° latitude × 2.5° longitude grid.

**2.3 The GEOS-Chem Model**

We use the GEOS-Chem global 3D chemical transport model version 13.0.2 (https://doi.org/10.5281/zenodo.4681204; last accessed May 2021) to represent contemporary understanding of upper tropospheric $NO_y$ for comparison to DC-8. The model is driven with consistent NASA Modern-Era Retrospective analysis for Research and Applications version 2 (MERRA-2) assimilated meteorology at 2° × 2.5° (latitude × longitude) over 47 vertical layers from the surface of the Earth to 0.01 hPa. The model emissions local

to the upper troposphere include cruising altitude aircraft from the Aviation Emissions Inventory Code (AEIC) (Stettler et al., 2011) and lightning emissions as described in Murray et al. (2012). Surface emissions of $NO_x$ and VOCs precursors of ALKNs and PNs are from the anthropogenic Community Emissions Data System (CEDS) inventory of Hoesly et al. (2018), the Model of Emissions of Gases and Aerosols from Nature (MEGAN) biogenic VOCs inventory version 2.1 (Guenther et al., 2012), the soil $NO_x$ emission inventory of Hudman et al. (2012),

and the Global Fire Emissions Database version 4 with small fires (GFED4s) for open burning of biomass (Giglio et al., 2013). Wet deposition of gas-phase $HNO_3$, the terminal sink for $NO_y$ subsiding from the upper troposphere, includes in-cloud (rainout) and above-cloud (washout) scavenging as detailed in Amos et al. (2012) and enhanced scavenging as described by Luo et al. (2020).

We sample the model at the same time and location as the DC-8 observations using the ObsPack diagnostic

(https://www.esrl.noaa.gov/gmd/ccgg/obspack/; last accessed 23 October 2024) following a minimum 10-month spin-up preceding each campaign to initialize chemistry and large-scale circulation throughout the troposphere. Modelled components of $NO_y$ include NO, $NO_2$, $HNO_3$, $HNO_4$, PAN, PPN, peroxymethacroyl nitrate (MPAN), MPN, and ALKNs.

**3 Results and Discussion**

**3.1 DC-8 campaign $NO_y$ seasonality and budget closure**

Figure 3 compares seasonality in UT $NO_y$ from IAGOS and DC-8. Most of the overlap is with ATom along the North Atlantic flight corridor in all seasons, ARCTAS over the Canadian Arctic and Greenland in March-May (MAM) and June-August (JJA), and SEAC$^4$RS over the Southeast US in September-November (SON). IAGOS $NO_y$ exhibits similar peaks in spring (563 pptv) and summer (565 pptv), due to intensive seasonal lightning in the



northern hemisphere (Stratmann et al., 2016). Decline in this source decreases NO$_y$ in autumn to 365 pptv and NO$_y$ further decreases in winter to an annual minimum of 284 pptv.

DC-8 NO$_y$ seasonality is similar to that of IAGOS, though the magnitude of DC-8 NO$_y$ is consistently on average ~130 pptv (range of 80 pptv in SON to 170 pptv in DJF) less than IAGOS NO$_y$ in all seasons. The ~130 pptv

greater IAGOS NO$_y$ likely results mostly from differences in sampling altitudes. The mean sampling altitude of IAGOS for all coincident 2° × 2.5° grid cells is ~240 hPa (~10 km), whereas the sampling altitude for DC-8 is on average ~1.5 km below that of IAGOS at ~360 hPa. According to the DC-8 measurements, NO$_y$ is ~100 pptv more at 240 hPa (IAGOS) than at 360 hPa (DC-8 mean altitude). Another minor factor may be IAGOS NO$_y$ instrument interference from HCN. The IAGOS chemiluminescence instruments use a hydrogen (H$_2$) reagent to

convert oxygenated nitrogen compounds to NO, whereas DC-8 uses CO, a compound not permitted on commercial aircraft (Bradshaw et al., 1998; Volz-Thomas et al., 2005; Thomas et al., 2015). The H$_2$ reagent converts anywhere from 2 to 20% of HCN to NO$_y$ (Weinheimer, 2006). HCN ambient concentrations typically seasonally vary from ~200 to 300 pptv in the upper troposphere, amounting to an interference of 4-60 pptv (Li et al., 2003; Le Breton et al., 2013).


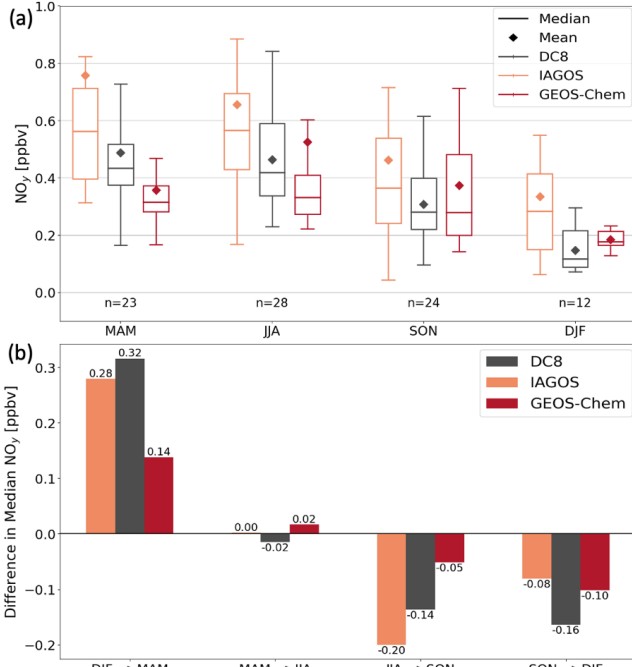

**Figure 3: Seasonality of northern hemisphere upper tropospheric NO$_y$. Panels show seasonal means and medians (a) and seasonal transitions (b) of collocated gridded 2° × 2.5° NO$_y$ from IAGOS (orange), DC-8**
**(grey), and GEOS-Chem (red). Data in (a) are medians (lines), 25th and 75th percentiles (boxes) and means (diamonds). Inset text in (a) gives the number (n) of overlapping grid cells. Seasonality in (b) is the change in median NO$_y$ in (a) from one season to the next.**



Figure 4 shows the relationship between the sum of individual $NO_y$ components and total $NO_y$ for each DC-8

campaign. We use these scatterplots to determine whether most $NO_y$ components are measured in each campaign, given our intention to use DC-8 to assess contemporary understanding of upper tropospheric $NO_y$. The individual components of $NO_y$ summed to compare to total $NO_y$ include NO; PSS $NO_2$ (Equation (1)); $HNO_3$; PAN measured as PAN for all ATom sub-campaigns and as part of total PNs for ARCTAS, SEAC⁴RS and KORUS-AQ; $HNO_4$ measured as $HNO_4$ for ATom-1 and -2 and as part of total PNs for ARCTAS, SEAC⁴RS and KORUS-AQ; C1-

C5 ALKNs for all AToms; total ALKNs for SEAC⁴RS, KORUS-AQ, and ARCTAS; PPN and other PANs for all except ATom-1 and -2; and MPN as part of total PNs for ARCTAS, SEAC⁴RS and KORUS-AQ. The evaluation in Figure 4 is biased toward the northern hemisphere, as the low time resolution sampling of ALKNs during ATom leads to loss of data in the southern hemisphere (Figure 2) to achieve coincidence of DC-8 total and individual components of $NO_y$.


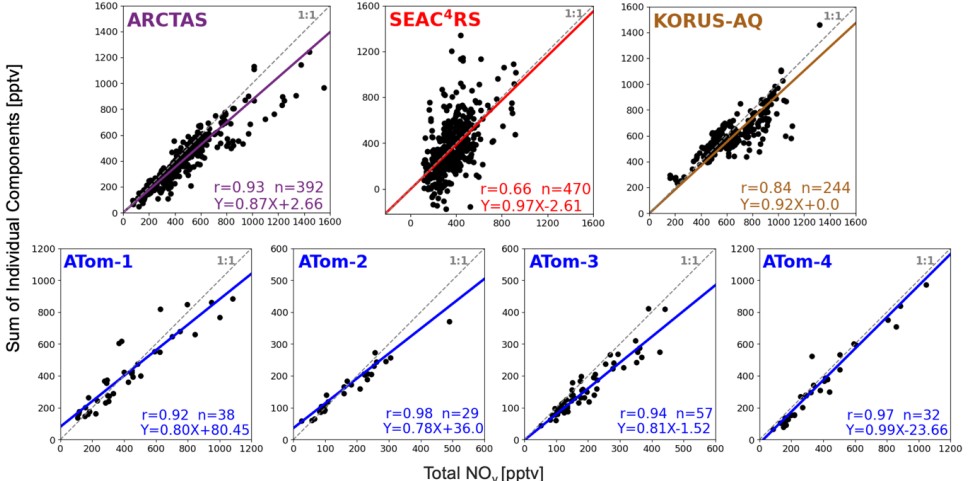

**Figure 4: Proportion of reactive nitrogen components measured during each campaign. Individual points compare the coincident sum of individual $NO_y$ components to measured total $NO_y$ during NASA DC-8 campaigns. Individual $NO_y$ components used in the figure are detailed in the text. Dashed grey lines are the**

**1:1 relationship. Coloured lines and inset equations are the Theil-Sen regression fit to the observations. Other inset values are the Pearson's correlation coefficient (r) and number of points (n). Axis ranges differ in each panel.**

Total measured $NO_y$ and the sum of individual $NO_y$ components are strongly correlated (r > 0.8) for all campaigns,

except SEAC⁴RS (r = 0.66). The weaker correlation for SEAC⁴RS is from the large contribution of MPN to total PNs measured by the TD-LIF instrument. If instead we replace TD-LIF PNs with the sum of CIMS PANs and $HNO_4$, the correlation with total measured $NO_y$ increases to r = 0.91, but the regression slope decreases from 0.97 in Figure 4 to 0.82, as MPN is ~20% of SEAC⁴RS $NO_y$. The large contribution of MPN to total $NO_y$ during SEAC⁴RS is from aged air initially influenced by lightning, biomass burning and deep convective uplift of surface

pollution with large amounts of VOCs and $NO_x$. These large amounts of VOCs and $NO_x$ cause very active photochemistry that enhances abundance of the MPN precursor, $CH_3O_2$ (Nault et al., 2015; Browne et al., 2011).



The regression slopes in Figure 4 indicate that most NO$_y$ components are measured during each campaign, ranging from 0.78 for ATom-2 (78% of individual NO$_y$ components measured) to 0.99 for ATom-4 (99% measured). The slopes suggest that between 1-22% of NO$_y$ originates either from unmeasured components, positive interference in the NO$_y$ instrument, or a combination of both. Bradshaw et al. (1998) estimated a temperature-dependent interference from HCN of 8-15% for chemiluminescence instruments that, like those deployed on DC-8 campaigns, use a CO reagent. We estimate a lower-end (8%) interference for mean ambient upper troposphere temperatures measured along the flight paths in Figure 2. Using DC-8 HCN observations, this amounts to ~53 ppt or 12% of NO$_y$ for ARCTAS, ~19 pptv or 5% of NO$_y$ for SEAC$^4$RS, ~40 pptv or 6% NO$_y$ for KORUS-AQ, and ~17 pptv or 6% of NO$_y$ for ATom 1-4.

**3.2 Upper tropospheric NO$_y$ composition**

Figure 5 provides a breakdown of the absolute and relative contributions of individual NO$_y$ components to total NO$_y$. ATom-1 and -4 are combined, as these sub-campaigns have a very similar range in NO$_y$ (Figure 4) and in median total and individual components of NO$_y$, as the sampled seasons (spring and summer) have very similar NO$_y$ (Figure 3). Similarly, ATom-2 and -3 (autumn and winter) are combined. Campaigns are further grouped into remote (ARCTAS, ATom) and continental (SEAC$^4$RS, KORUS-AQ), as local influence from continental sources like anthropogenic emissions and intense lightning leads to a greater relative contribution of NO$_x$ and lesser contribution of PAN for the continental upper troposphere and vice versa for the remote upper troposphere.

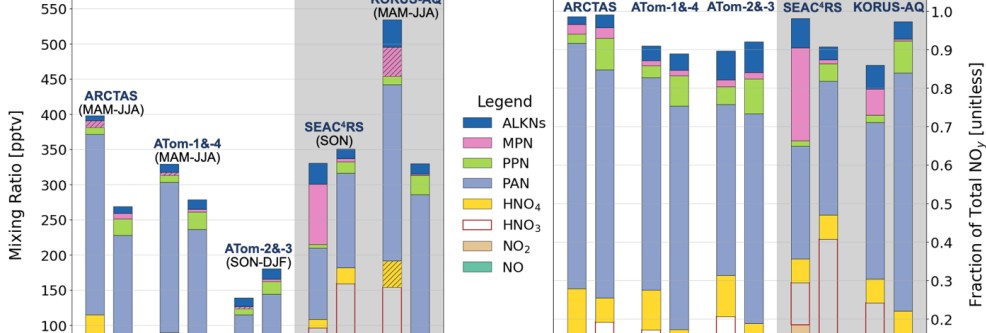

**Figure 5: NO$_y$ composition in the upper troposphere along DC-8 flight tracks. Bars are median values of absolute (a) and relative (b) individual NO$_y$ components observed and inferred from the observations during DC-8 campaigns and simulated by GEOS-Chem (GC). Seasons sampled are given above each bar (a) and the grey shading distinguishes sampling in the remote (no shading) and continental (shaded) upper troposphere. Hatchings in (a) indicate inferred concentrations (see text for details). Bar components from bottom to top are NO, NO$_2$, HNO$_3$, HNO$_4$, PAN, PPN, MPN, and ALKNs.**

DC-8 quantities in Figure 5 not directly measured are inferred. These include ATom-3 and -4 HNO$_4$, ATom-1 and -2 PPN, ARCTAS MPN, KORUS-AQ HNO$_4$ and MPN, and ATom MPN. We use ATom-1 HNO$_4$ and ATom-



4 PPN for combined ATom-1 and -4 components, and, similarly, ATom-2 $HNO_4$ and ATom-3 PPN for combined ATom-2 and -3. ARCTAS upper tropospheric MPN of ~10 pptv is from the estimate by Browne et al. (2011). KORUS-AQ $HNO_4$ is estimated to be 37 pptv by multiplying the SEAC⁴RS median fraction of $HNO_4$ ($HNO_4/NO_y$

= 0.06) by the KORUS-AQ median $NO_y$. SEAC⁴RS is used, as $HNO_4$ is thermally unstable (Ryerson et al., 2000) and so varies with temperature. Mean upper troposphere ambient temperatures for KORUS-AQ (252 K) are more consistent with SEAC⁴RS (246 K) than the other campaigns (238 K for ARCTAS, 238K-241 K for ATom).

KORUS-AQ MPN can be estimated by bounding a potential range from two approaches. The first is the median

value of the difference between TD-LIF total PNs and the sum of all individual CIMs PANs and our inferred $HNO_4$, yielding MPN = 75 pptv. This likely overestimates MPN, as the CIMS instrument does not measure an exhaustive suite of PANs. Lee et al. (2022) estimated with a box model and KORUS-AQ measurements that unmeasured PANs account for ~20% of total PNs during KORUS-AQ, though this is for air masses impacted by petrochemical and other anthropogenic VOCs and $NO_x$ emissions. Accounting for these unmeasured PANs yields

a lower-bound KORUS-AQ MPN of 8 pptv. The MPN that we infer then for KORUS-AQ is 42 pptv, the midpoint of 8 and 75 pptv, accounting for 7% of KORUS-AQ $NO_y$. As the GEOS-Chem model MPN is consistent with DC-8 inferred MPN during ARCTAS, we multiply the GEOS-Chem ATom MPN fractions ($MPN/NO_y$ ~0.01 for ATom-1 and -4 and ~0.02 for ATom-2 and -3) by ATom DC-8 $NO_y$ to infer ATom MPN.

It is challenging to infer concentrations of longer chain (>C5) ALKNs during ATom, so only the C1-C5 ALKNs are shown in Figure 5. The remote measurements of total ALKNs available from ARCTAS that would be most suitable to use to infer ATom total ALKNs are on median 5 pptv less than the ATom C1-C5 ALKNs measurements and are very noisy, as indicated by a range of -113 pptv to ~333 pptv. The range in ARCTAS WAS C1-C5 measurements, by comparison, is 8-29 pptv. Contributions of >C5 ALKNs to total ALKNs for SEAC⁴RS (~50%)

and KORUS-AQ (~60%), representative of the continental upper troposphere, can only be used to suggest that >C5 ALKNs in remote regions are <50% of total ALKNs or <12 pptv (median of C1-C5 ALKNs for ATom1-4). According to the measurements, remote region C1-C5 ALKNs are dominated by methyl nitrate (C1 ALKN), accounting for 40% of ATom C1-C5 ALKNs and 49% for ARCTAS. Second is isopropyl nitrate (C3 ALKN), making up 17% of ATom C1-C5 ALKNs and 25% for ARCTAS. The >C3 ALKNs dominate ALKNs in the

continental upper troposphere, accounting for 92% of total ALKNs for SEAC⁴RS and 71% for KORUS-AQ. These we estimate as the difference between TD-LIF total ALKNs and the sum of WAS C1-C3 ALKNs.

The sum of KORUS-AQ $NO_y$ components total 531 pptv, >130 pptv more than SEAC⁴RS, ARCTAS, and ATom-1 and -4 that are within a narrow range of 330-400 pptv. Minimum $NO_y$ are for the remote autumn and winter

measurements from ATom-2 and -3 at 141 pptv. Despite the wide range in absolute total and components of $NO_y$, the relative contribution of many individual $NO_y$ components is consistent across all campaigns. These include NO (7 ± 3%; mean ± 1σ standard deviation), $NO_2$ (6 ± 2%), $HNO_3$ (9 ± 2%), $HNO_4$ (9 ± 3%), PPN (3 ± 1%), and ALKNs (5 ± 3%). PAN, the dominant $NO_y$ component in all campaigns, is least consistent, ranging from 30-41% for the continental upper troposphere to 44-64% for the remote upper troposphere. The $HNO_4$ fraction (10-13%)

in the remote upper troposphere is more than the continental upper troposphere (~6%), due to colder temperatures



for ATom and ARCTAS. MPN is uniquely prominent during SEAC⁴RS, accounting for 24% of NO$_y$ compared to 2-7% inferred for all other campaigns.

The NO$_y$ composition information in Figure 5 has a northern hemisphere sampling bias to achieve coincidence. ATom observations south of the Equator exhibit a similar seasonal pattern to the northern hemisphere: summer > spring > autumn > winter NO$_y$, except that the southern hemisphere spring and summer NO$_y$ differ by ~90 pptv, whereas there is near-negligible difference for the northern hemisphere (Figure 3). As with the northern hemisphere, PAN accounts for most southern hemisphere NO$_y$, ranging from ~32% for ATom-1 (July-August) to ~42% for ATom-2 (January-February).


Measurements of the NO$_y$ compounds nitrogen pentoxide (N$_2$O$_5$), the gas-phase nitrate radical (NO$_3$), and nitrous acid (HONO) are not included in Figure 5, as these are prominent at night and have near-negligible daytime abundances. Of these, there are only measurements of N$_2$O$_5$, limited to ATom-3 and -4, that represent ~0.1% of upper tropospheric NO$_y$ along the daytime ATom flight tracks in Figure 2. NO$_3$ has a lifetime of a few seconds

during the day, due to rapid photolysis and prompt reaction with NO (Brown and Stutz, 2012). HONO rapidly photolyzes with a near-surface lifetime of 15 min (Sörgel et al., 2011) that would be much shorter in the upper troposphere where photolysis frequencies are enhanced.

**3.3 Contemporary understanding of UT NO$_y$**


GEOS-Chem northern hemisphere upper troposphere NO$_y$ is compared to the observations in Figures 3 and 5. In Figure 3, GEOS-Chem median NO$_y$ is less than DC-8 in summer and spring by ~103 pptv, similar to DC-8 in autumn, and greater than DC-8 in winter by ~60 pptv. As a result of these differences in absolute NO$_y$, the model underestimates the IAGOS and DC-8 seasonal shifts in NO$_y$ from winter to spring and from summer to autumn.


The sum of the GEOS-Chem fractional contributions of NO$_y$ components in Figure 5(b) that do not sum to 1 are because the model NO$_y$ budget also includes components not measured during DC-8, such as MPAN and halogenated ALKNs. Consistent across all campaigns is model underestimate in NO$_2$ and overestimate in PPN. The model version we use does not include photolysis of PPN, even though this is known to occur (Harwood et

al., 2003). PPN photolysis rather than thermal decomposition is the dominant loss pathway of PPN in the cold upper troposphere. PPN photolysis is scheduled for inclusion in a later model version (version 14.5) than is used here (Horner et al., 2024). Inclusion of PPN photolysis would liberate up to ~16 pptv NO$_2$, resolving the 10-16 pptv model underestimate in NO$_2$. Other studies have addressed model biases in NO$_2$ by including photolysis of aerosol nitrate (pNO$_3$) forming HONO that rapidly photolyses to NO$_x$ (Shah et al., 2023; Horner et al., 2024).

pNO$_3$ concentrations are too small in the upper troposphere for this to be a substantial NO$_2$ source. Aerosol Mass Spectrometer (AMS) measurements of pNO$_3$ are on median ~0.01 µg m$^{-3}$ during ARCTAS, ~0.07 µg m$^{-3}$ during KOUS-AQ, ~0.04 µg m$^{-3}$ during SEAC⁴RS and <0.01 µg m$^{-3}$ during ATom.

The model exhibits significant campaign-specific biases in total NO$_y$ for ARCTAS (129 pptv underestimate),

KORUS-AQ (205 pptv underestimate), ATom-1 and -4 (51 pptv underestimate) and ATom-2 and -3 (42 pptv





overestimate). The model underestimate in ARCTAS $NO_y$ is due mostly to a ~100 pptv low bias in PAN and, to a lesser extent, a 35 pptv underestimate in $HNO_4$. The model bias for ATom-2 and -3 is due almost entirely to PAN. For KORUS-AQ, all $NO_y$ components except PPN are underestimated, indicative of an overall underestimate in reactive nitrogen sources to the upper troposphere over this region. The ATom-1 and -4 underestimate in $NO_y$ is due mostly to a low model bias in PAN and $HNO_3$. Overall, the model underestimates the contrast in upper tropospheric $NO_y$ between the remote and continental upper troposphere.


GEOS-Chem simulates individual C1-C3 ALKNs, but most >C3 ALKNs are included as a lumped species. There are other >C3 ALKNs represented individually in the model, such as those formed from isoprene oxidation (Fisher et al., 2016), but abundances of these are near-negligible in the upper troposphere. DC-8 C1 ALKN is only 4% of ALKNs for SEAC$^4$RS and 11% for KORUS-AQ, whereas in the model these are a much greater component of ALKNs: 40% for SEAC$^4$RS and 29% for KORUS-AQ. Modelled >C3 ALKNs are a far smaller portion of total ALKNs (29% for SEAC$^4$RS and 23% for KORUS-AQ) than the observations (Sect. 3.2). Modelled C1 ALKN concentrations are consistently less than the observed values by ~2 pptv for ARCTAS and ~1 pptv for ATom. Modelled C3 ALKN is ~1 pptv less than the observations for ARCTAS, but ~1 pptv more than the observations for ATom.



The sum of measured and modelled individual $NO_y$ components are not significantly different for SEAC$^4$RS, though the model overestimates $HNO_3$ by 64 pptv and underestimates MPN by 81 pptv. The model low bias in MPN suggests that the model underestimates influence of $NO_x$ and reactive VOCs sources on aged air over source regions with a mix of emissions from fires and lightning, and deep convective injection of surface pollution. The model high bias in $HNO_3$ could be because of a factor of 2 model overestimate in $H_2O_2$ compared to observed $H_2O_2$ for SEAC$^4$RS. An overestimate in $H_2O_2$ indicates a model overestimate in $HO_2$ that promotes formation of $HNO_3$ and that would also account for the ~10 pptv overestimate in modelled $HNO_4$. Modelled $HO_2$ is used to calculate PSS $NO_2$ for SEAC$^4$RS (Equation (1), Sect. 2.1), but this only imparts a small high bias (~1.7 pptv) in SEAC$^4$RS PSS $NO_2$. Model bias in $H_2O_2$ for ARCTAS (>100 pptv) may also be the cause for the model underestimate in ARCTAS $HNO_4$ of ~35 pptv.



Modelled KORUS-AQ $HNO_3$, ALKNs, and MPN are all biased low. The low biases in these $NO_y$ components may be because of a general underestimate in $NO_y$ sources over South Korea where there are large anthropogenic $NO_x$ and VOCs sources that are represented in the model with a global inventory (CEDs) that may not suitably account for local emissions (Travis et al., 2024). Lightning $NO_x$ emissions could also be underestimated in the heavily parameterized inventory in GEOS-Chem (Murray et al., 2012; Marais et al., 2018), but this is a challenging $NO_x$ source to evaluate over locations that include other prominent sources of $NO_x$.



The model biases identified in this work hinder accurate determination of the radiative effect of tropospheric ozone for short-term climate impact assessments, the oxidative capacity of the troposphere for quantifying the lifetime and persistence of the greenhouse gas methane, tropospheric column densities of $NO_2$ from space-based UV-visible instruments that are retrieved with modelled vertical profiles of $NO_2$, $NO_x$ emissions by comparing



modelled and observed oxidized nitrate wet deposition fluxes that depend on the abundance of soluble $HNO_3$, and harm of reactive nitrogen deposition to vulnerable habitats.

**4 Conclusions**

We used NASA DC-8 aircraft measurement data from the ARCTAS, SEAC$^4$RS, KORUS-AQ, and ATom campaigns to characterize reactive nitrogen ($NO_y$) in the global upper troposphere. This followed confirmation from comparison to routine reactive nitrogen measurements from the IAGOS commercial aircraft campaign that DC-8 has the same seasonality of peak $NO_y$ in summer and spring and minimum $NO_y$ in winter in the northern hemisphere. Consistency supports use of DC-8 campaign data to characterise $NO_y$ under standard daytime

conditions.

We also confirm that most (78-99%) $NO_y$ components were measured during DC-8 campaigns. These include nitrogen oxides ($NO_x$), and inorganic ($HNO_3$ and $HNO_4$), and organic (PANs, MPN, and alkyl nitrates) reservoirs of $NO_x$. PAN is the dominant $NO_y$ component for all campaigns (30-64%), followed by $NO_x$ (6-18%), $HNO_4$ (6-

13%) and $HNO_3$ (7-11%). The relative contribution of most other components is similar across all campaigns, except for MPN that is ~24% of $NO_y$ for SEAC$^4$RS over the southeast US and much less (2-7%) for all other campaigns, though the latter range is from inferred concentrations of MPN.

The GEOS-Chem model is sampled along the DC-8 flight tracks to assess the state of knowledge of upper

tropospheric $NO_y$. Consistent model biases for all campaigns include an overestimate in PPN and underestimate in $NO_2$. The model lacks PPN photolysis that would address the PPN model bias and mostly resolve the $NO_2$ bias. In the continental upper troposphere, the model underestimates total $NO_y$ for KORUS-AQ, but reproduces total $NO_y$ for SEAC$^4$RS, though with too much $HNO_3$ and too little MPN. Over remote regions, the model biases are less severe, and are likely related to the weak seasonal variability in total $NO_y$ in comparison to DC-8 and IAGOS.

A possible cause of this is errors in model representation of maritime lightning $NO_x$ emissions that influence $NO_y$ abundance in spring and summer.

Our results underscore the need for sustained measurements of upper tropospheric reactive nitrogen for further refinement of knowledge of upper tropospheric $NO_y$ sources, advection, and chemical processing. This is crucial

for advancing our understanding of the global nitrogen cycle and its broader environmental implications.

**Author Contributions**
Study concept by EAM and NW. NW led the data analysis and simulated GEOS-Chem. The manuscript is initiated by NW and co-written with EAM. GL aided in data analysis, RGR in the use of ObsPack, and BS in the use of

IAGOS $NO_y$ observations. All authors reviewed and edited the manuscript.

**Competing interests**
The authors declare that they have no conflict of interest.



**Acknowledgements**

We are grateful for the provision of the NASA DC-8 aircraft observations provided by the instrument PIs Paul O. Wennberg, Ronald C. Cohen, Thomas B. Ryerson, Chelsea Thompson, Andrew Weinheimer, L. Gregory Huey, Jim Elkins, and Donald R. Blake and, for IAGOS, the IAGOS-Core data provided by Andreas Volz-Thomas and IAGOS-CARABIC by Helmut Ziereis. The authors acknowledge the strong support of the European Commission, Airbus, and the airlines (Lufthansa, Air France, Austrian Airlines, Air Namibia, Cathay Pacific, Iberia and China Airlines so far) who have carried the IAGOS-Core equipment and performed the maintenance since 1994. IAGOS-CARABIC $NO_y$ measurement funding is from the German Aerospace Centre (DLR). In its last 10 years of operation, IAGOS-Core has been funded by INSU–CNRS (France), Météo-France, Université Paul Sabatier (Toulouse, France) and Forschungszentrum Jülich (FZJ, Jülich, Germany). IAGOS has been additionally funded by the EU projects IAGOSDS and IAGOS-ERI. The IAGOS-Core and the IAGOS-CARIBIC database are supported by AERIS. IAGOS-CARIBIC data are also available from the IAGOS-CARIBIC team (see http://www.caribic-atmospheric.com )

**Data and Software Availability**

All data and software used in this study are from publicly accessible repositories cited in the text.

**Funding**

This research has been supported by the European Research Council under the European Union's Horizon 2020 research and innovation programme (through a Starting Grant awarded to Eloise A. Marais, UpTrop [grant no. 851854]).

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
