# Peer review of "Characterization of reactive oxidized nitrogen in the global upper troposphere using recent and historic commercial and research aircraft campaigns and GEOS-Chem"

_EGUsphere, 2024_

## Author Comment (AC1)

**RESPONSE TO REVIEWERS**

Ms. Ref. No.: egusphere-2024-3388, doi:10.5194/egusphere-2024-3388

Journal: Atmos. Chem. Phys.

Point-by-point responses to reviewers are included below. Reviewer comments are in blue. Responses are in black, and text added or altered is quoted in orange. Line numbers are those in the updated tracked changes PDF document.

**Responses to Reviewer RC1:**

*Wei et al. characterize the distribution, seasonality, and speciation of reactive oxidized nitrogen ($NO_y$) in the upper troposphere using measurements from research campaigns and commercial flights and compare them to a global model simulation to test our understanding of $NO_y$ sources & chemistry. They analyze the similarities and differences in the $NO_y$ speciation in different regions and seasons and identify discrepancies between the observations and the model. This study tackles an important topic since the cycling of oxidized nitrogen in the global troposphere affects tropospheric ozone and OH, and thus climate and tropospheric oxidant levels. The paper is very well-written and the conclusions, for the most part, are well supported by the data and analysis presented. Here are some points that need to be clarified further:*

*(1) The UT is defined as 8-12 km. This is appropriate for the mid-latitudes. but not the tropics. I understand this might be because the DC8 has a ceiling of 12 km, but this should be clarified.*

We now clarify this in Section 2.1:

"The DC-8 research aircraft has sampled ambient air covering the near full extent of the troposphere" (line 150)

"... within a wide pressure range from 180 hPa (~8 km) to the DC8 ceiling of 450 hPa (~12 km). This captures the full vertical extent of the midlatitude upper troposphere, but not the tropics. The tropical tropopause, according to NASA Modern-Era Retrospective analysis for Research and Applications version 2 (MERRA-2) meteorology, extends to ~16 km." (lines 176-179)

*(2) Lines 159-185: The authors use $NO_2$ calculated from the photochemical steady state instead of the measurements because of interference in the chemiluminescence instrument. The SEAC4RS TD-LIF $NO_2$ measurements may also be biased high (Silvern et al. 2018, Shah et al. 2023) and this could affect the TD-LIF measurements of the sum of PANs, etc. which are*

*calculated by subtracting the measured NO₂. Is a correction to TD-LIF measurements of these NOy species needed?*

The largest contributor to TD-LIF $NO_2$ interference is MPN (100% decomposition; Reed et al. (2016)). Shah et al. (2023) estimate a 0-21% $NO_2$ interference for SEAC⁴RS TD-LIF $NO_2$ measurements. If we use 21% as an upper bound, this is ~5 pptv of the median SEAC⁴RS PSS $NO_2$; a small contribution (~3%) to the 190 pptv PNs.

We adjust the text to include it in the discussion of Figure 4:

"... from factors such as unmeasured components, positive interference in the $NO_y$ instrument, or a low bias in the TD-LIF PNs ..." (Lines 384-385)

We now discuss this in the paper in the context of Figure 4:

"TD-LIF measurements of PNs are calculated from the difference in $NO_2$ detected with the $NO_2$ channel and with the PNs channel set to a temperature at which all PNs decompose (Day et al., 2002). A bias in $NO_2$ could therefore impart a bias in PNs. The largest source of TD-LIF interference is 100% decomposition of MPN (Reed et al., 2016) and MPN during SEAC⁴RS far exceeds any of the other campaigns. If we use the higher-end interference of 21% from Shah et al. (2023) for SEAC⁴RS, this equates to ~5 pptv of SEAC⁴RS PSS $NO_2$. This is only ~3% of the 190 pptv SEAC⁴RS PNs." (lines 399-404)

*(3) One of the paper's conclusions is that GEOS-Chem underestimates MPN during SEAC⁴RS. MPN seems to make up an unexpectedly large fraction of the observed $NO_y$ during SEAC⁴RS, and including it in the sum of the $NO_y$ species degrades the correlation with the total $NO_y$ measurements (Figure 4). This leads me to suspect that the MPN measurements may be biased high. It would be valuable if the authors could dig in a little more to find further support for the MPN measurements, and the conclusion that our understanding of MPN sources is rather poor. They could, for example, look at the ATom data downwind of the southeast US to see if there is a substantial difference between the total $NO_y$ and the sum of the $NO_y$ species. Do other studies show a significant underestimate of VOCs in GEOS-Chem during SEAC⁴RS?*

The ATom measurements are much closer to North Africa (Figure 2 in the main manuscript) than to the southeast US, so would not be suitable for corroborating the large MPN contribution to total $NO_y$ during SEAC⁴RS. If we instead infer MPN by subtracting the sum of $HNO_4$ and

all PANs measured with the CIMS instrument from the TD-LIF PNs measurement and assume most PANs are measured by the CIMS instrument, MPN declines from 85 to 49 pptv (from 24% to 14% of NO) for SEAC$^4$RS. This is 32 pptv less than the MPN in Figure 5. Even so, MPN still makes a greater relative contribution to total NO$_y$ during SEAC$^4$RS than is inferred for the other campaigns and far more than is simulated with GEOS-Chem.

This is added to Section 3.2:

"The far larger fraction of MPN to total NO$_y$ during SEAC$^4$RS (Figure 5(b)) warrants further investigation. If we instead estimate MPN by subtracting the sum of HNO$_4$ and all PANs measured with the CIMS instrument from the TD-LIF PNs, making the assumption that CIMS measures most PANs, MPN is 49 pptv and the contribution to NO$_y$ declines from 24% to 14%. This is still at least double the contribution for any other campaign." (Lines 483-486)

And incorporated in the comparison to the model in Section 3.3 to confirm that a substantial model underestimate in MPN still holds:

"The sum of measured and modelled individual NO$_y$ components are not significantly different for SEAC$^4$RS, though the model overestimates HNO$_3$ by 64 pptv and underestimates MPN by 81 pptv compared to the TD-LIF measurements and by 45 pptv compared to MPN inferred using TD-LIF PNs and CIMS HNO$_4$ and PANs (Section 3.2)." (Lines 562-565)

The abstract and conclusions are also updated to incorporate this range in MPN contribution and to give the most conservative model bias:

"Methyl peroxy nitrate (MPN) makes an outsized contribution to NO$_y$ (14-24%) over the Southeast US." (Lines 21-22)
"A model underestimate in MPN of at least ~50 pptv (13-fold) over the Southeast US" (Lines 25-26)
"The relative contribution of most other components is similar across all campaigns, except for MPN that is 15-24% of NO$_y$ for SEAC$^4$RS over the Southeast US and much less (2-7%) for all other campaigns, though MPN measurements are rare and susceptible to biases." (Lines 605-607)

The potential underestimate in modelled VOCs measured during SEAC⁴RS is confirmed by a persistent low model bias calculated by Chen et al. (2019b) for free tropospheric VOCs of ~60%. We now include this in Section 3.3:

"Chen et al. (2019a) estimated that the GEOS-Chem underestimate in free tropospheric VOCs during SEAC⁴RS is on average ~60%, but exceeds a factor of 2 for many of the VOCs assessed." (Lines 567-568)

*(4) Lines 54 – 75: I presume nighttime $NO_y$ chemistry in the UT is slow enough to be ignored, but it would be good to describe it briefly and state why it is not important, if that is the case.*

We now describe the dominant nighttime chemistry, acknowledge its importance and the severe lack of detailed measurements to aid our understanding:

"Nighttime $NO_y$ chemistry is also important, but aircraft campaign measurements of the nocturnal upper troposphere are mostly of total $NO_y$ from commercial aircraft campaigns. The nighttime chemistry not in Figure 1 includes NO reaction with OH forming nitrous acid (HONO) that accumulates in the absence of photolysis, as well as $NO_2$ reaction with $O_3$ to form the nitrate radical ($NO_3$) that further reacts with $NO_2$ to produce $N_2O_5$, a precursor of aerosol nitrate (pNO₃) (Bradshaw et al., 2000)." (p. 2 lines 90-92; p. 3 lines 100-101)

*(5) Lines 365- 372: The paragraph discusses the minor $NO_y$ species not included in the analysis, but it does not discuss particulate nitrates (organic & inorganic). Shouldn't these aerosol species be included in $NO_y$, even if they are minor in the UT?*

We discuss the potential contribution of pNO3 to the upper troposphere $NO_y$ budget in the context of Figure 4 that, for 100% sampling efficiency, 4% of total $NO_y$:

"Chemiluminescence $NO_y$ instruments also measure pNO₃, but with uncertain sampling efficiencies (Bourgeois et al., 2022). For 100% efficiency and using the Aerosol Mass Spectrometer (AMS) measurements of submicron (< 1 μm) pNO₃, the contribution is at most 1% of $NO_y$ for ARCTAS for a median pNO₃ of ~0.01 μg m⁻³ (~4 pptv), ~4% for SEAC⁴RS for pNO₃ of ~0.04 μg m⁻³ (~14 pptv), ~4% for KORUS-AQ for pNO₃ ~0.07 μg m⁻³ (~25 pptv), and <2% for ATom for pNO₃ <0.01 μg m⁻³ (~4 pptv)." (lines 393-397)

And again in the context of Figure 5:

"pNO$_3$, absent in Figure 5 due to the uncertain sampling efficiency of the chemiluminescence instrument, is at most 4% for SEAC$^4$RS and for KORUS-AQ (Section 3.1), comparable to the contribution from PPN." (lines 479-481)

*(6) I suggest that NO$_y$ be called "reactive oxidized nitrogen" instead of "reactive nitrogen," which includes reduced nitrogen (NH3, etc.). And also that in this work NO$_y$ does not include N$_2$O.*

Thank you for the suggestion. We have replaced all instances of "reactive nitrogen" with "reactive oxidized nitrogen".

**References:**

Bourgeois, I., Peischl, J., Neuman, J. A., Brown, S. S., Allen, H. M., Campuzano-Jost, P., Coggon, M. M., DiGangi, J. P., Diskin, G. S., Gilman, J. B., et al.: Comparison of airborne measurements of NO, NO$_2$, HONO, NO$_y$, and CO during FIREX-AQ, Atmos. Meas. Tech., 15, 4901-4930, doi:10.5194/amt-15-4901-2022, 2022.

Bradshaw, J., Davis, D., Grodzinsky, G., Smyth, S., Newell, R., Sandholm, S., and Liu, S.: Observed distributions of nitrogen oxides in the remote free troposphere from the NASA Global Tropospheric Experiment Programs, Rev. Geophys., 38, 61-116, doi:10.1029/1999rg900015, 2000.

Chen, X., Millet, D. B., Singh, H. B., Wisthaler, A., Apel, E. C., Atlas, E. L., Blake, D. R., Bourgeois, I., Brown, S. S., Crounse, J. D., et al.: On the sources and sinks of atmospheric VOCs: An integrated analysis of recent aircraft campaigns over North America, Atmos. Chem. Phys., 19, 9097-9123, doi:10.5194/acp-19-9097-2019, 2019a.

Chen, X., Millet, D. B., Singh, H. B., Wisthaler, A., Apel, E. C., Atlas, E. L., Blake, D. R., Bourgeois, I., Brown, S. S., Crounse, J. D., et al.: On the sources and sinks of atmospheric VOCs: an integrated analysis of recent aircraft campaigns over North America, Atmos. Chem. Phys., 19, 9097-9123, doi:10.5194/acp-19-9097-2019, 2019b.

Day, D. A., Wooldridge, P. J., Dillon, M. B., Thornton, J. A., and Cohen, R. C.: A thermal dissociation laser-induced fluorescence instrument for in situ detection of NO$_2$, peroxy nitrates, alkyl nitrates, and HNO$_3$, J. Geophys. Res.: Atmos., 107, doi:10.1029/2001jd000779, 2002.

Reed, C., Evans, M. J., Di Carlo, P., Lee, J. D., and Carpenter, L. J.: Interferences in photolytic NO$_2$ measurements: explanation for an apparent missing oxidant?, Atmos. Chem. Phys., 16, 4707-4724, doi:10.5194/acp-16-4707-2016, 2016.

Shah, V., Jacob, D. J., Dang, R., Lamsal, L. N., Strode, S. A., Steenrod, S. D., Boersma, K. F., Eastham, S. D., Fritz, T. M., Thompson, C., et al.: Nitrogen oxides in the free troposphere: implications for tropospheric oxidants and the interpretation of satellite $NO_2$ measurements, Atmos. Chem. Phys., 23, 1227-1257, doi:10.5194/acp-23-1227-2023, 2023.

---

## Author Comment (AC2)

**RESPONSE TO REVIEWERS**

Ms. Ref. No.: egusphere-2024-3388, doi:10.5194/egusphere-2024-3388

Journal: Atmos. Chem. Phys.

Point-by-point responses to reviewers are included below. Reviewer comments are in blue. Responses are in black, and text added or altered is quoted in orange. Line numbers are those in the updated tracked changes PDF document.

**Response to Reviewer RC2:**

*Wei et al. use different aircraft measurements combined with a global model simulation to characterise upper tropospheric NOy and assess our understanding of the processes governing it. The manuscript is generally well written and easy to follow. The topic is important and timely, affecting for example tropospheric oxidation capacity and ozone formation.*
*Below, I have listed certain areas where I would still like to see more detail, followed by a list of minor comments.*

*1. The comparison between the IAGOS and DC-8 flights. There is a rather large difference in NOy levels between the two, which is explained to result from differences in flight altitudes (lines 241-242). As the comparison of these two flight measurements forms a core part of the manuscript, I would like to see more detailed comparisons here, like altitude profiles of NOy from the two types of measurements. Do they match up?*

We now include a supplementary figure (Figure S1; pasted below) showing the seasonal mean vertical profiles of collocated DC-8 and IAGOS $NO_y$ and we discuss the features in this figure in the text to support the distinct altitude ranges sampled by DC-8 and IAGOS and consistency in $NO_y$ for the few instances that sampling is vertically collocated.

"The two campaigns sample distinct altitude ranges of the upper portion of the upper troposphere centred at ~240 hPa (~10 km) for IAGOS and a wider vertical extent of the lower portion of the upper troposphere centred at ~360 hPa (~1.5 km below IAGOS) for DC-8 (Figure S1). There is a general pattern of a steep increase in $NO_y$ with altitude, with the exception of IAGOS layers located near 300 hPa in March-May and September-November (Figure S1). Average $NO_y$ is similar between the two campaigns for the rare instances that DC-8 and IAGOS sample the same pressure layers (Figure S1)." (lines 316-322)

[Figure]

Figure S1: Comparison of seasonal mean vertical profiles of total reactive nitrogen ($NO_y$) from spatially collocated DC-8 and IAGOS aircraft observations. Symbols are means from averaging upper troposphere (450-180 hPa) observations into 30 hPa bins. Lines are standard deviations. Shading indicates the typical vertical sampling range (pressure standard deviation) of DC8 (grey) and IAGOS (orange). Pressure range selection and screening for stratospheric influence and plumes are detailed in the main manuscript.

*2. SEAC$^4$RS stands out from the other campaigns in Figs. 4 and 5. These differences are expected to arise from the high contribution of MPN to NOy in that campaign. Especially the poor correlation seen in Fig. 4 leads me to suspect there may be something wrong with these measurements, or then that MPN would not be properly reflected in the NOy measurements. Could you analyse further, whether these MPN measurements are indeed high and correct, or may there be some interference in them?*

Further analysis assessing a potential high bias in MPN is detailed in the first part of the response to reviewer # 1, Comment (3). Even with this smaller MPN concentration, the conclusion that the model substantially underestimates MPN still holds.

*3. Altitude definitions: upper troposphere is here defined as 8-12 km in altitude. However, tropopause may be kilometers higher in the tropics: can you justify the choice of altitude range further?*

This is addressed in response to Reviewer #1, Comment (1).

*Minor comments:*
*Abstract, lines 23-24: fractional/percentage values of the over/underestimation would be useful here*

Added as 10-90% for PPN and 31-65% for $NO_2$. (lines 23-24)

*Line 54, Fig. 1: can you provide references justifying these are the main species & reactions?*

To better represent the key studies that have informed dominance of $NO_y$ components and the reactions in Figure 1, we have edited the text and cited these studies as below:

"Chemical cycling of dominant daytime $NO_y$ components, informed by past review and measurement compilation studies of the free troposphere (Emmons et al., 1997; Bradshaw et al., 2000), is illustrated in Figure 1." (lines 73-74).

*Lines 61-63: add ref for the photolysis vs thermal decomposition*
Huey (2007) added. (Line 82)

*Lines 65-68: add refs*
Schultz et al. (1999) added for $NO_x$ reservoir compound transport and subsidence. (Lines 88-87)

Text edited to clarify that existing citations substantiate slow loss processes in the upper troposphere:

"Loss processes in the dry upper troposphere are slow and dominated by subsidence, resulting in long $NO_y$ lifetimes of 10-20 days (Logan, 1983; Prather and Jacob, 1997). Similarly, $NO_x$

has a lifetime of about a week compared to less than a day in the boundary layer (<2 km) (Jaeglé et al., 1998)." (Lines 87-89)

*Lines 76-78: add ref*

We have edited the text, so that the citations appear before the sentence starting "These studies have either focused on ...". For completeness, we have also added other relevant citations. The updated text reads as:

"Modelling studies evaluating best understanding of $NO_y$ in the upper troposphere routinely identify stark discrepancies between observed and modelled total $NO_y$, $NO_x$, and the ratio of NO-to-$NO_2$ in the upper layers of the troposphere (Jaeglé et al., 1998; Talbot et al., 1999; Bertram et al., 2007; Hudman et al., 2007; Liang et al., 2011; Nault et al., 2015; Huntrieser et al., 2016; Travis et al., 2016; Fisher et al., 2018; Silvern et al., 2018; Lee et al., 2022; Cohen et al., 2023). These studies have either focused on a few $NO_y$ components, or a single aircraft campaign." (Lines 107-112)

*Line 104: add ref*

Singh et al. (2006) added. (Line 155)

*Lines 105-107: mention that the exact definitions for screening will follow*

Thank you for the suggestion. We rather decided to delete "... and that have limited influence from stratospheric air", as this data screening aspect was not a factor in eliminating INTEX-NA and DC3. (Line 156)

*Line 126: would times relative to sunrise and sunset be more appropriate? You screen for jNO2 as well, but this leads to different representation of high- and low latitudes (as mentioned on lines 139-140). I'm not requesting you to redo all the analyses, but preferably comment on if this has an effect on the results*

We do already use the reported solar zenith angle (SZA) values to confirm that are time range and $jNO_2$ filter removes data with large SZA (sunrise/sunset). (Line 201). Given this, we do not expect any effect on our results if we instead used SZA.

*Line 127: full vertical extent. But this does not include full vertical extent, esp. in tropics*

This text has been updated to address the similar Reviewer #1, Comment (1).

*Screening criteria: how much data do these criteria exclude? In other words, how typical are the sought-for background conditions?*

We now state provide these statistics and the main cause for data loss:

"The proportion of observations at 450-180 hPa is 42-50% for ATom and 16-37% for the other campaigns. After applying all other data screening, 20% of all data are retained for ATom and 7-11% for the other campaigns." (Lines 202-203).

*Lines 135-136: refs for these screening criteria*

These are already provided as Hudman et al. (2007) and as Shah et al. (2023). (Lines 196-198)

*Line 137: what does approximately zero mean?*

Zero within the range of uncertainty of the instrument.

*Line 149: ref talks about TD-CIMS specifically*

Thank you for pointing this out. We have replaced the Slusher et al. (2004) reference with Huey (2007). (Line 212)

*Lines 151-153: ref*

This information is obtained directly from the dataset.

*Line 154: ref for TD-LIF*

Day et al. (2002) added. (Line 217)

*Eq. 1, also in the text: NO and NO2 should be in square brackets*

These are intentionally not in square brackets, as the units are distinct (pptv) from the compounds in squares brackets (molecules/cm$^3$).

*Line 181: how is HO2 measured?*

$HO_2$ is measured using a laser induced fluorescence.

*Line 184, RO2 relatively insignificant: I could not easily find this in the reference. Is it so?*

Shah et al. (2023) state that it makes a small contribution in the free troposphere, so we have reworded our text to more closely match theirs:

"... but we ignore this reaction as it is relatively insignificant throughout the free troposphere (Shah et al., 2023)." (Lines 248-249).

*Line 202: are O3, CO and jNO2 also measured on the commercial aircraft?*

$O_3$ and CO are, as is now made clear in the text:

"... and daytime filtering as is applied to DC-8 data (Sect. 2.1) using IAGOS $O_3$ and CO measurements." (Line 273).

There are no $jNO_2$ measurements, but there is no need to include this filtering step, as none of the coincident flights extend to the high latitudes. Now also stated in the text:

"There are no $NO_2$ photolysis frequency measurements, but the requirement for spatial coincidence with ATom excludes polar twilight and night measurements at high latitudes." (Lines 274-275)

*Line 222: do you mean below-cloud?*

Yes. Corrected (Line 293).

*Lines 242-243: ref or show data*

Data are now shown in response to your Comment 1.

*Fig. 3 b: is panel b needed?*

Yes. The panel helps illustrate the size of seasonal variability in total $NO_y$. Despite large differences in absolute concentrations of DC8 and IAGOS $NO_y$, both exhibit relatively similar seasonal changes, whereas the model (state of knowledge) seasonal shifts in $NO_y$ are too modest.

*Lines 261-266: would it be easier to read if common measurements were listed, and then campaign-wise which compounds were included?*

We now summarise this information in a new Table 1 (pasted below) for greater clarity and refer to the table in the Figure 1 caption and in the text (Line 350).

**Table 1.** Observations of individual $NO_y$ components summed to assess budget closure in Figure

| Component | NASA DC-8 aircraft campaign | | |
|---|---|---|---|
| | ARCTAS, SEAC⁴RS, KORUS-AQ | ATom1-2 | ATom3-4 |
| $NO_2$ | PSS | PSS | PSS |
| NO | Chemiluminescence (CL) | CL | CL |
| $HNO_3$ | CIMS | CIMS | CIMS |
| $HNO_4$ | TD-LIF PNs | CIMS | – |
| PAN | TD-LIF PNs | PANTHER | PANTHER |
| PPN | TD-LIF PNs | – | CIMS |
| other PANs | TD-LIF PNs | – | CIMS |
| ALKNs | TD-LIF ALKNs | WAS C1-C5 | WAS C1-C5 |
| MPN | TD-LIF PNs | – | – |

*Lines 294-296: does including the HCN observations improve the agreement?*

At the risk of over-interpreting a rough estimate of HCN interference, we now state that the HCN interference estimates are consistent with the unaccounted $NO_y$ in Figure 4:

"These lower-end interference estimates are similar in size to the percent missing $NO_y$ (13% for ARCTAS, 3% for SEAC⁴RS, 8% for KORUS-AQ, 1-22% for ATom)." (Lines 390-391)

*Section 3.2: includes long and complicated section on inferred concentrations, should this rather be in the methods section?*

We prefer to keep it in this section, as the inference requires knowledge of the median values of the measured $NO_y$ components presented in Figure 5.

We now state in the methods Section 2.1 that inference is needed for quantities not measured:

"The $NO_y$ components not measured during specific campaigns are inferred. These include $HNO_4$ for KORUS-AQ, and ATom-3-4, PPN for ATom-1-2, and MPN for ARCTAS, ATom-1-4 and KORUS-AQ. The approaches used to infer these values differs, informed by the results, so a detailed description of this inference is in Section 3.2." (Lines 251-254)

And in Section 3.2, we adjust the text of the 2 paragraphs to accommodate this pre-introduction to inference in the methods section and to focus paragraph 1 on HNO4 and PPN and paragraph 2 on MPN:

Lines 425-442:
"Inferred DC-8 $HNO_4$ and PPN in Figure 5 use ATom-1 $HNO_4$ and ATom-4 PPN for combined ATom-1 and -4 components, and, similarly, ATom-2 $HNO_4$ and ATom-3 PPN for combined ATom-2 and -3. KORUS-AQ $HNO_4$ is estimated to be 37 pptv by multiplying the SEAC⁴RS median fraction of $HNO_4$ ($HNO_4/NO_y = 0.06$) by the KORUS-AQ median $NO_y$. SEAC⁴RS is used, as $HNO_4$ is thermally unstable (Ryerson et al., 2000) and so varies with temperature. Mean upper troposphere ambient temperatures for KORUS-AQ (252 K) are more consistent with SEAC⁴RS (246 K) than the other campaigns (238 K for ARCTAS, 238K-241 K for ATom)."

The inferred ~10 pptv ARCTAS MPN is from the estimate by Browne et al. (2011). KORUS-AQ MPN is estimated by bounding a potential range from two approaches. The first is the median value of the difference between TD-LIF total PNs and the sum of all individual CIMs PANs and our inferred $HNO_4$ of 37 pptv, yielding MPN = 75 pptv. This likely overestimates MPN, as the CIMS instrument does not measure an exhaustive suite of PANs. Lee et al. (2022) estimated with a box model and KORUS-AQ measurements that unmeasured PANs account for ~20% of total PNs during KORUS-AQ, though this is for air masses impacted by petrochemical and other anthropogenic VOCs and $NO_x$ emissions. Accounting for these unmeasured PANs yields a lower-bound KORUS-AQ MPN of 8 pptv. The MPN that we infer then for KORUS-AQ is 42 pptv, the midpoint of 8 and 75 pptv, accounting for 7% of KORUS-AQ $NO_y$. As the GEOS-Chem model MPN is consistent with DC-8 inferred MPN during ARCTAS, we multiply the GEOS-Chem ATom MPN fractions (MPN/$NO_y$ ~0.01 for ATom-1 and -4 and ~0.02 for ATom-2 and -3) by ATom DC-8 $NO_y$ to infer ATom MPN of < 6 pptv."

We have also rewritten the first half of the ALKNs paragraph to focus it on the results in Figure 5, rather than the challenges of inferring >C5 ALKNs during ATom:

"Only the C1-C5 ALKNs are shown in Figure 5 for ATom. The remote measurements of total ALKNs available from ARCTAS that would be most suitable to assess the likely contribution of longer chain (>C5) ALKNs are on median 5 pptv less than the ATom C1-C5 ALKNs measurements. The total ARCTAS total ALKNs measurements are also very noisy, as indicated by a range of -113 pptv to ~333 pptv. The range in ARCTAS WAS C1-C5 measurements, by comparison, is 8-29 pptv. Contributions of >C5 ALKNs to total ALKNs for SEAC$^4$RS (~50%) and KORUS-AQ (~60%), representative of the continental upper troposphere, suggest that >C5 ALKNs in remote regions are <50% of total ALKNs or <12 pptv (median of C1-C5 ALKNs for ATom1-4)." (p. 12 lines 443-447; p.13 lines 461-463)

*Line 359: coincidence of the individual to total NOy? What about disregarding ALKN as they make a relatively small fraction of NOy to achieve higher southern hemisphere coverage?*

Indeed, this approach could have been adopted instead, but as we do already include a paragraph dedicated to the southern hemisphere (Lines 488-493) that covers the relative

contribution of the dominant component PAN, and the seasonality in total $NO_y$ that we find to be generally consistent with the northern hemisphere biased findings.

*Lines 366-368: ref for negligible contribution*

These are already provided following the initial opening sentence of the paragraph. Following that sentence, we elaborate that $NO_3$ and HONO have very short lifetimes, as both rapidly photolyze.

*Line 370: check the reference. For 100 ppt NO the lifetime is 15 seconds, so for the max average NO campaign (SEAC4RS) it is close to a minute. Noontime clear sky photolysis lifetime is around 6 s, longer off-noon and at high latitudes. Not saying that NO3 would be significant, but that people often underestimate NO3 daytime lifetimes*

Thank you for pointing out this issue. We have amended the text:

$NO_3$ has a lifetime of a few seconds during the day, due to efficient photolysis (Brown and Stutz, 2012). (Lines 497-498)

*Lines 371-372: how much shorter?*

We now state this in terms of the 50% faster photolysis rates expected for HONO from knowledge of the wavelength at which is photolyzes and the increase in actinic flux and hence photolysis frequencies with altitude at this wavelength:

Photolysis of HONO would be further enhanced (by ~50% at 390 nm) in the upper troposphere where photolysis frequencies are enhanced (Hofzumahaus et al., 2002; Reed et al., 2016). (p. 13, line 499; p. 14, lines 514-515)

*Lines 391-392: ref, or is this your result?*

Our result. This is clarified in updated text added to address Reviewer #1, Comment (5). To accommodate that update, we amend the text in these lines too:

"These are on median, ~0.01 µg m$^{-3}$ during ARCTAS, ~0.07 µg m$^{-3}$ during KORUS-AQ, ~0.04 µg m$^{-3}$ during SEAC$^4$RS and <0.01 µg m$^{-3}$ during ATom (Section 3.1)." (Lines 533-534)

*Line 417: ref*

Text slightly modified to clarify that this is our analysis of the data:

"The model high bias in HNO$_3$ could be because of a factor of 2 overestimate in our modelled H$_2$O$_2$ compared to observed H$_2$O$_2$ for SEAC$^4$RS." (Lines 569-580)

*Lines 460-461: was this mentioned in the results section?*

Yes. In Section 3.1, we discuss the role of lightning in affecting seasonality of total NO$_y$ measured by IAGOS and DC8. (Lines 310-311)

*Line 490: maybe cite all sources here again.*

Done. (Lines 651-653)

**References:**

[revised manuscript text omitted]

---

## Author Response (AR2)

**RESPONSE TO EDITOR**

Ms. Ref. No.: egusphere-2024-3388, doi:10.5194/egusphere-2024-3388

Journal: Atmos. Chem. Phys.

Point-by-point responses to remaining editor comments are given below. Comments are in blue and responses in black. Text added or altered is quoted in orange.

**Responses to Editor comments:**

*The authors have done an excellent job responding to the comments. A couple of minor thoughts to further improve clarification would further clarify some of the comments by the reviewers:*

*1) An SI figure that either colors the scatter plot of NOy from SEAC4RS either as MPN mixing ratio or the fraction of MPN to total NOy, or the difference of the sum and measured NOy vs MPN may help with better understanding the worst agreement.*

We now include Figure S2 showing the SEAC⁴RS panel from Figure 4 and the points coloured by the relative contribution of MPN to total $NO_y$:

[Figure]

**Figure S2: Proportion of reactive oxidized nitrogen components measured during SEAC⁴RS. Figure format and inset values are as in Figure 4, but for SEAC⁴RS only and points are coloured by the relative proportion of methyl peroxy nitrate (MPN).**

And we draw the reader's attention to the figure in Section 3.1 in the paragraph immediately following Figure 4:

"The weaker correlation for SEAC$^4$RS is from the large contribution of MPN to total PNs measured by the TD-LIF instrument, leading to a large contribution of MPN to total NO$_y$ for many of the points that stray most from the 1:1 line (Figure S2)."

We also reference Figure S2 as motivation for further exploring the large contribution of MPN to total NO$_y$ during SEAC$^4$RS:

"The far larger fraction of MPN to total NO$_y$ during SEAC$^4$RS (Figure 5(b)) warrants further investigation, as the relative proportion of MPN to total NO$_y$ ranges from negligible to 100% (Figure S2)." (page 13)

*2) There is potential that PNA interference in MPN channel may also not be fully accounted for. A brief discussion and or quick look if the discrepancy could be due to double counting of PNA from the CIMS and TDLIF may be warranted here for further clarification.*

We now use the ~11% HNO$_4$ interference from Nault et al. (2015) to calculate a non-significant impact of HNO$_4$ interference on MPN during SEAC$^4$RS:

"A small proportion of HNO$_4$ is measured in the MPN channel of the TD-LIF instrument. About 11%, according to Nault et al. (2015). For CIMS median HNO$_4$ of 12.6 pptv during SEAC$^4$RS, HNO$_4$ interference is only 1.4 pptv, so does not affect the 14-24% contribution." (page 13)

**Reference:**

Nault, B. A., Garland, C., Pusede, S. E., Wooldridge, P. J., Ullmann, K., Hall, S. R., and Cohen, R. C.: Measurements of $CH_3O_2NO_2$ in the upper troposphere, Atmos. Meas. Tech., 8, 987-997, doi:10.5194/amt-8-987-2015, 2015.